# A Novel Approach to Condition Monitoring of the Cutting Process Using Recurrent Neural Networks

**DOI:** 10.3390/s20164493

**Published:** 2020-08-11

**Authors:** Rui Silva, António Araújo

**Affiliations:** COMEGI, Campus de Vila Nova de Famalicão, Universidade Lusíada–Norte, Edifício da Lapa-Largo Tinoco de Sousa, 4760-108 Vila Nova de Famalicão, Portugal; antonio.araujo@hotmail.com

**Keywords:** recurrent neural networks, condition monitoring, tool wear

## Abstract

Condition monitoring is a fundamental part of machining, as well as other manufacturing processes where, generally, there are parts that wear out and have to be replaced. Devising proper condition monitoring has been a concern of many researchers, but there is still a lack of robustness and efficiency, most often hindered by the system’s complexity or otherwise limited by the inherent noisy signals, a characteristic of industrial processes. The vast majority of condition monitoring approaches do not take into account the temporal sequence when modelling and hence lose an intrinsic part of the context of an actual time-dependent process, fundamental to processes such as cutting. The proposed system uses a multisensory approach to gather information from the cutting process, which is then modelled by a recurrent neural network, capturing the evolutive pattern of wear over time. The system was tested with realistic cutting conditions, and the results show great effectiveness and accuracy with just a few cutting tests. The use of recurrent neural networks demonstrates the potential of such an approach for other time-dependent industrial processes under noisy conditions.

## 1. Introduction

Condition monitoring is present throughout all industry activities and plays a major role in process efficiency, being increasingly integrated in machine tools in an effort towards unmanned machining. Suitable condition monitoring of manufacturing processes is fundamental to the success of other prominent activities, such as tool change policies, production control and operations management in general. Among all manufacturing activities, the ones where materials are shaped by material removal, such as turning or milling operations, require special attention, since it is a complex process conditioned by a significant number of variables such as cutting parameters, tool and machined part materials and machine characteristics, to name a few [1,2,3]. A key strategy to support the goal of unmanned machining is to develop sensor-based, real-time monitoring systems that can aid in classifying tool wear levels and promote adequate tool change policies.

Condition monitoring can be subdivided into several stages: sensor selection and deployment, generation of a feature indicative of tool condition and, finally, classification (i.e., assessing the collected and processed information to determine the level of wear on the tool). This process is often perceived as an attempt to mimic human sensory behavior in that it resembles the processing of different sensed information into an intelligent decision-making process. In tool wear monitoring, many sensors and features have been considered so far (e.g., vibration, sound, temperature, force and many other sources of information) [4,5,6,7]. Supported by previous research results [7,8,9] and the literature [1,2], it was found that the best candidates for condition monitoring of the cutting process were force, vibration and sound, being the most feasible and least intrusive sensors that relate more accurately to the wear process of tools. In addition, the use of multiple sensors should prove to be of great value towards tool wear evaluation, since the noisy characteristics of data captured by each sensor alone would lead to certain failure of the monitoring system [5,6,9]. Although condition monitoring has been extensively studied, some limitations persist concerning the performance of monitoring systems under realistic machining conditions [10]. Condition monitoring systems have been proposed using many different approaches, some with different success rates and limitations, such as neural networks [3,7,11,12,13,14], fuzzy logic [15,16], Markov chains [17,18,19], expert systems [20] and many others [10]. Neural networks present some of the most attractive features, such as the capability of abstraction of hardly accessible knowledge and generalization from distorted sensor signals when applied to sensor fusion and classification in tool wear monitoring. The use of artificial intelligence is common, given its flexibility and the fact that there are universal nonlinear approximators outperforming direct modelling through mechanistic approaches [21].

Computational models for neural systems have often concentrated on the processing of static stimuli using typical feedforward neural networks, as reported in the vast majority of research on condition monitoring using artificial intelligence [11,14]. However, numerous biologically relevant signals have a rich temporal structure, and neural circuits must process these signals in real time and hence give context to temporal sequences, such as in audition, where almost all information is embedded in the temporal structure. Furthermore, in the visual domain, movement represents one of the fundamental features extracted by the nervous system; hence, it is not surprising that, in the last few years, there has been an increased interest in the dynamic aspects of neural processing. The processing of real-world, time-varying stimuli is a difficult problem and represents a challenge for artificial models of neural functions [11,22]. This paper explores the feasibility of using recurrent neural networks (RNNs) in condition monitoring of the cutting process. This novel approach takes advantage of the underlying characteristics of RNNs that demonstrate unique capacities in modelling dynamic complex systems and sequenced information [23,24,25]. A brief introduction to RNNs is provided, highlighting major aspects concerning the base algorithm, architecture and proposed learning policy that supports the monitoring strategy and justifies its feasibility and suitability. Experimental work is detailed afterwards to support the evaluation of the current approach. Finally, simulation results are presented that demonstrate the flexibility and feasibility of RNNs for tool condition monitoring of the cutting process.

## 2. Recurrent Neural Networks

Recurrent neural networks possess a certain number of characteristics that make them unique, namely the capabilities associated with traditional neural networks, as well as inherent sequence capturing capabilities, making them especially suited to dealing with time series and time-dependent problems [26]. In theory, an RNN can capture information pertaining to arbitrarily long sequences and hence incorporate information pertaining to time-dependent events in its memory or model. Recurrent neural network principles were first proposed in the 1980s [27,28,29] and set the background for the development of recent frameworks and variations [30]. It should be noticed that, as in Figure 1, the initial network topology is unfolded through time, with the different weighted connections being shared at all times. The distinctive feature of this neural network is that hidden weights between time steps propagate forward past information, therefore conditioning future results while building a temporal structure.

A recurrent neural network (RNN) computes the hidden vector sequence activation H = {H_0_, H_1_, …, H_T_} and the output sequence O = {O_0_, O_1_, …, O_T_}, given the input sequence X = {X_0_, X_1_, …, X_T_}. X_i_ represents the input feature vector at a given time step, extracted from acquired sensor data; W represents the hidden weight matrix; U is the forward weight from the X inputs; and V is the weight matrix from the hidden units to the output nodes. Both the hidden and output sequences are related through the following equations, for each time step:(1)$$H_{t}=\tanh\left(U\cdot X_{t}+W\cdot H_{t-1}\right)$$
(2)$$O_{t}=V\cdot H_{t}$$

As in traditional neuronal networks, weights must be adjusted to perform adequate classifications and hence learn from examples. One of the most used algorithms to gradually adjust weights in an RNN resembles the backpropagation algorithm and was named backpropagation through time (BPTT) [31]. As in traditional backpropagation algorithms, there is a need to determine the gradient of errors to enable gradual adjustments on the different weights [27]. Upon the presentation of a given time-sequenced input and output, learning is performed at each time step layer of the unfolded network, starting from the last time step. The fundamental difference from the backpropagation algorithm is related to the fact that hidden weights are also responsible for the unfolding of the network and must be adjusted at every iteration of the unfolded network structure. Given the need to repeatedly recalculate error gradients over time, it causes gradient error values to expand or shrink exponentially, leading to the problem of vanishing and exploding gradients [32]. These problems are more likely to occur when deeper unfolding is required, such as in learning long time sequences, but can be superseded through second-order optimization techniques such as Hessian-free optimization [33] or modifying the network structure, giving rise to solutions resembling a long short-term memory (LSTM) network, featuring multiplicative gates so that values can be retained over time [30,34]. Weight adjustments are performed in all weight matrices (U, V and W) using the backpropagation algorithm and calculated using the chain rule of differentiation. Since the weights W are used at every time step, it is necessary to backpropagate all gradients and, finally, update these weights, as in the traditional backpropagation algorithm. As errors are summed up, so too are error gradients added up for each training set while unfolding the RNN.

### Architecture and Learning Strategy

The success of applying recurrent neural networks depends on many factors, such as network structure, learning strategy and the ability to overcome the vanishing and exploding gradients problem [31]. The choice of a given architecture depends on the problem that is being dealt with and may vary depending on the adopted approach. For instance, when learning to model a time series, the network can be fed with successive inputs that can go as deep as one expects it to capture information, depending upon its behavior, while the output can be set to a single neuron, being a one step ahead or multiple steps ahead prediction. This same problem could be handled with time-differed paired input–output samples unfolding as deep as required, giving rise to one or multiple steps ahead predictions. Tool wear in cutting is inherently a time-dependent process and hence a suitable problem to be dealt with using recurrent neural networks. The information captured from the sensors at each time step should be fed to the unfolding network and, as a result, an estimate for tool wear level should be provided by the network at each time step. Tool wear monitoring demands that classification is performed gradually at each time step, meaning that the unfolding of the recurrent neural network should be performed, and learning carried out, in a different fashion from traditional recurrent neural networks. The unfolding depth remains small for the limited number of tool wear sampled data for each cutting insert, and hence vanilla recurrent neural networks can be used without any extra precaution regarding vanishing and exploding gradients.

The recurrent neural network developed to support the presented results was coded based on the principles introduced by Hopfield (1982) [29] and Rumelhart et al. (1986) [27]. Considerations regarding the used architecture and learning strategy were inspired by past research, mainly in the review by Schmidhuber [31]. The network consists of three layers: an input layer of 14 inputs corresponding to the different features extracted from the sensor data, a hidden layer and one output neuron, corresponding to the tool wear level of a given set of features from the sampled sensor data. While performing classification, the network unfolds to a given depth, corresponding to the sequence of sampled data, from new tools to the current wear stage. While monitoring the cutting process, the network progressively unfolds, providing a classification of tool wear for each new data set while retaining previously sensed data, cumulatively contributing to each new wear level classification. The importance of past data is intuitively important because the wear process is a progressive mechanism that builds upon past events. Input feature vectors are normalized so that different feature scales do not interfere with meaningful results, and weights are initialized to small random values so that map unfolding occurs smoothly, allowing the vanishing and exploding gradient effects to occur. In the early stages of training, feature vectors are presented for each cutting tool, limiting the temporal sequence; hence, only a few wear levels are initially shown to the network so that it starts to encode the temporal signature of the cutting process. During the last stages of training, the full temporal sequence of cutting tools is presented to refine learning and further encode all stages of a worn tool. This learning strategy allows for a gradual capture of the temporal structure inherent to tool wear progression, as well as limiting any threatening exponential degradation of weight gradients.

## 3. Materials and Methods

To test the feasibility of the proposed method, experiments were conducted in a test stand, based on a lathe equipped with force sensors; feed and tangential force measurements, through strain gauges placed in the tool holder; sound emission, with a microphone placed near the cutting tip; and an accelerometer, placed on the base of the machine. Figure 2 depicts the overall layout.

### 3.1. Experimental Procedure

Data were acquired while machining a block of mild steel with a coated, cemented carbide tip under the following cutting conditions: cutting speed of 350 m/min; feed rate of 0.25 mm/rev; and a depth of cut of 1 mm. Sampling of raw sensor data was conducted at a sampling frequency of 20 kHz on all sensors, with analogue prefiltering at 10 kHz to avoid aliasing. Data were acquired every 2 min, considering an expected tool life of approximately 15 min, with the tool flank wear being measured each time. Each sample consisted of 512 points for each sensor. Test data were collected from 6 tool tips at approximately 80 mm from both ends, in the middle of the workpiece. The workpiece, a bar initially 75 mm in diameter and 173 mm long, was held at one end by a special 250 mm diameter, 3-jaw, type 87 international power chuck and supported at the other end by a motorized programmable tailstock unit.

### 3.2. Experimental Results and Feature Extraction

A total of 14 features were extracted for each record at each of the tool wear levels. The extracted features for both sound and vibration, each of these sensors accounting for 6 features, were as follows: mean value, standard deviation, kurtosis, skewness and the power in the frequency bands 2.2–2.4 kHz and 4.4–4.6 kHz. Additionally, two more features were obtained from the mean value of the tangential and feed forces. All features exhibited a strong noise influence that could be associated with different factors, such as magnetic interference, tool and material impurities and machine dynamics. The data presented in Figure 3, relative to the evolution of tool wear in 6 different tool tips, are elucidative of the variability and unpredictability associated with monitoring tool wear, having different tools follow slightly different tool wear trajectories. Typically, tool wear takes place in three stages. The first stage is a short period of rapid wear. The wear then progresses at a slower rate over a period in which most of the useful tool life lies. The last stage is a rapid period of accelerated wear, and it is usually recommended that the tool be replaced before this stage, at about a value of flank wear (Vb) of 0.3 mm.

The tangential force (Figure 4) showed an overall consistent correlation with time, although different tools behaved differently. In addition, it can be seen that for most tools, the recorded value of the tangential force did not increase at every time step. In some cases, the recorded values of the average tangential force dropped slightly, which might be due to the complexity of the cutting process and noise induced by the industrial-like environment. Figure 5 shows the evolution of the skewness of sound against time, and a random-like behavior can be observed. The mean value, standard deviation, skewness and kurtosis of sound and vibration signals, the remaining features, also showed little correlation to tool wear when considered one at a time. Despite the random-like behavior, these features carry valuable information pertaining to the evolution of cutting dynamics due to tool wear, and they are not transparent to traditional mathematical modelling techniques [9,35].

As such, these sensors and features must be combined to take advantage of their embedded information, since each of the sensors alone carries a lot of noise and depicts a complex behavior. The complex relationship between different features, as well as the underlining noise, does not encourage the use of traditional mechanistic approaches. Hence, the use of neural networks, given their generalization and modelling capabilities, adds up to the unique characteristics of recurrent neural networks in capturing sequences.

## 4. Simulation Results and Discussion

The simulation was conducted with different hidden layer sizes, training lengths and learning rates. All weights were initialized to small values, between 0 and 0.01, to avoid vanishing or exploding gradients, although these were unlikely to happen given the short time sequence. The data from the first five cutting tools were used to train the recurrent neural network, and the last tool tip was used to assess the ability of the network to classify unseen data. During training, data pertaining to one of the tools were taken randomly from the training set and fully presented in the correct time sequence to the network, letting it unfold until the last tool wear level, as shown in Figure 6. Training was performed upon full unfolding in order to update the weights at all different levels. The present approach copes automatically with different sample sizes, since it unfolds recursively until the last acquired tool wear state (N). This network’s fitness for online condition monitoring is hence an intrinsic ability and suitable for all kinds of time-sequenced events.

The classification average mean error for the tip of Tool 4 (the dataset used for training) was 0.0007 mm, and the classification average mean error for the tip of Tool 6 (the unseen dataset) was 0.014 mm, with results presented in Figure 7. The average mean error of 0.0007 mm for the training set (previously learned information) was obtained from the difference between the measured and classification values. The average mean error of 0.014 mm for the unseen dataset was obtained from the measured and predicted values. Hence, the classification performance on the training data stays within 0.2% of the maximum value for the tool wear, and the unseen tool tip was, on average, classified to a precision of 3.5% of the maximum value. The classification of previously seen data was accurate, recalling to a high precision what had been learned. The unseen data also achieved a very good performance, considering that they had not been previously seen by the network and, therefore, the network proved to be able to capture the underlying behavior of the wearing process and still retain its generalization capabilities.

The best performance results were attained with a learning rate coefficient of 0.2, 5 × 10^4^ batch training cycles and a size of 8 neurons in the hidden layer. Figure 8 shows the effect of the learning rate upon classification of unseen data, depicting an optimum value of 0.2 around which learning takes place more efficiently, giving rise to a smaller average classification mean error. Increasingly smaller learning rates tended to degrade the learning capacity of the network. Learning rates above 0.6 disabled the network’s capacity to learn. For learning rates above 0.6, the error gradients within the network exploded and hence weights tended towards zero, leading to an average mean error corresponding to the mean of all tool wear classification. Figure 8 and Figure 9 outline the performance of the network with regards to different learning rates and different hidden layer size.

From Figure 9, it can be seen that the best performance upon classification of unseen data was achieved with a hidden layer size between 6 and 9 neurons. With a lower number of neurons, the network showed difficulties in capturing the overall system’s dynamics and hence classification degenerated, giving a higher average mean error. With a higher number of neurons in the hidden layer, classification degraded, probably due to difficulties in generalizing from the presented data and most certainly overfitting the learned patterns. Figure 10 further explores the performance of the network concerning the sensitivity to the number of training cycles.

Figure 10 shows the performance of the network when the number of training cycles is increased. It can be seen that with only 5000 cycles, the network converged rapidly to lower average mean errors, and further training had a small impact on performance. Learning was not disrupted with further training, and hence the captured relationship was preserved and performance was held with no overfitting.

## 5. Conclusions

This paper describes the design, implementation and training strategy of a prototype tool wear monitoring system based on recurrent neural networks. The design takes advantage of the inherent capabilities of these neural networks to capture sequences and thus reveals itself as a perfect modelling candidate for the characteristic progressive wear of tools. The results are very promising, since classification of unseen data is performed with a very low average mean error. This seemingly simple approach can be extended to other monitoring processes apart from machining, since it reveals flexibility and good generalization capabilities. Further work should be developed to test the network’s ability to perform one step ahead predictions, using an additional neuron in the unfolded output layer. In addition, further experiments should be conducted using smaller time intervals between sampling in order to assess the impact on performance and robustness of the proposed approach.

## Figures and Tables

**Figure 1 sensors-20-04493-f001:**
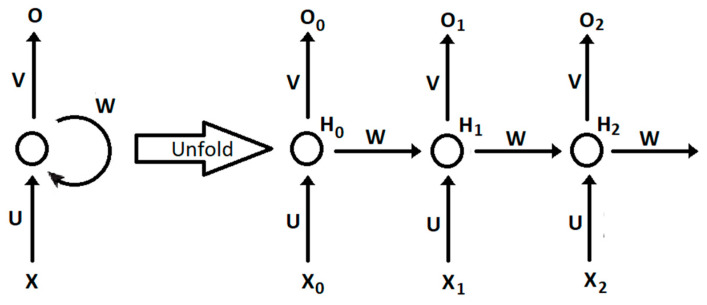
The unfolding of a recurrent neural network.

**Figure 2 sensors-20-04493-f002:**
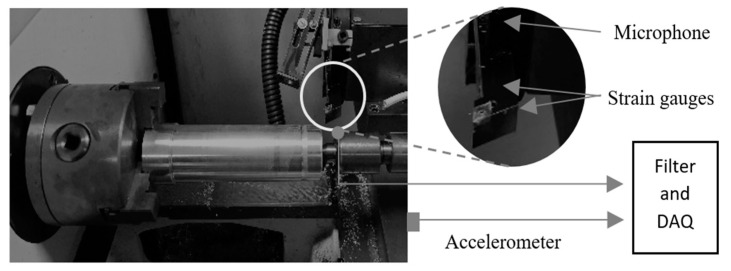
The experimental apparatus.

**Figure 3 sensors-20-04493-f003:**
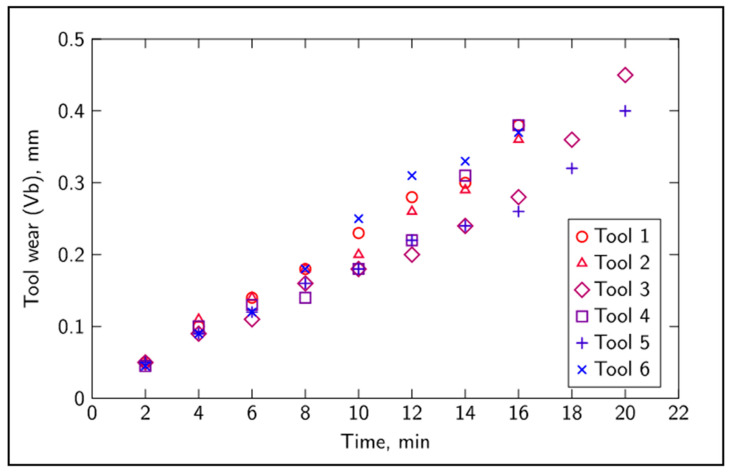
Flank wear evolution over time for all tools.

**Figure 4 sensors-20-04493-f004:**
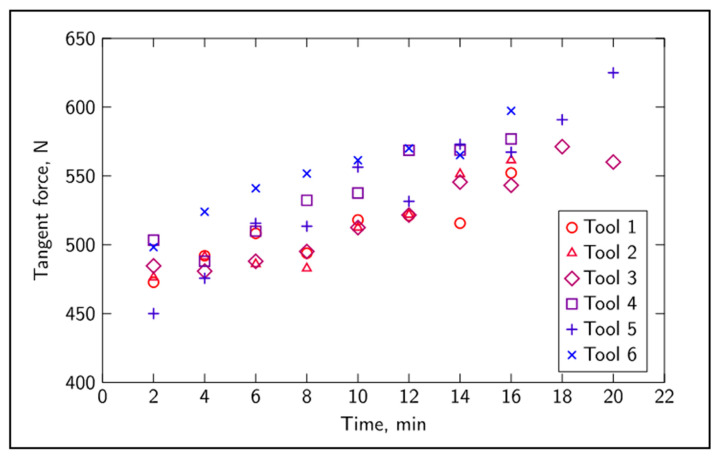
Tangential force evolution over time for all tools.

**Figure 5 sensors-20-04493-f005:**
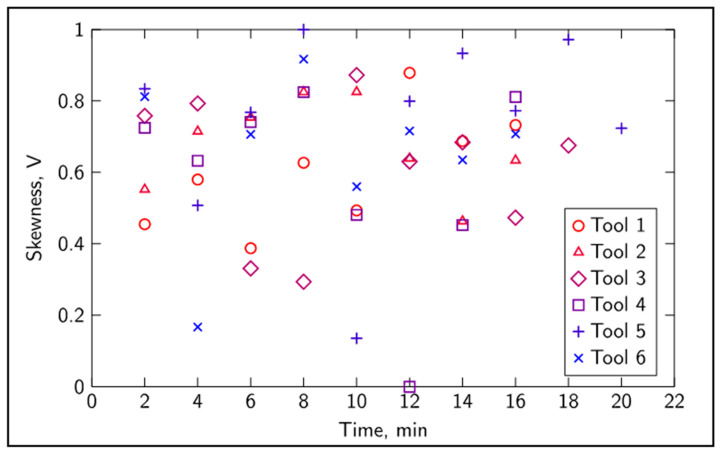
Normalized skewness evolution of sound over time for all tools.

**Figure 6 sensors-20-04493-f006:**
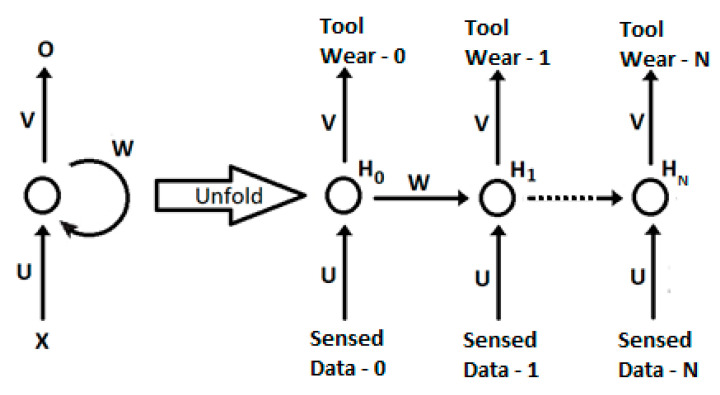
Unfolding of the recurrent neural network for tool wear classification.

**Figure 7 sensors-20-04493-f007:**
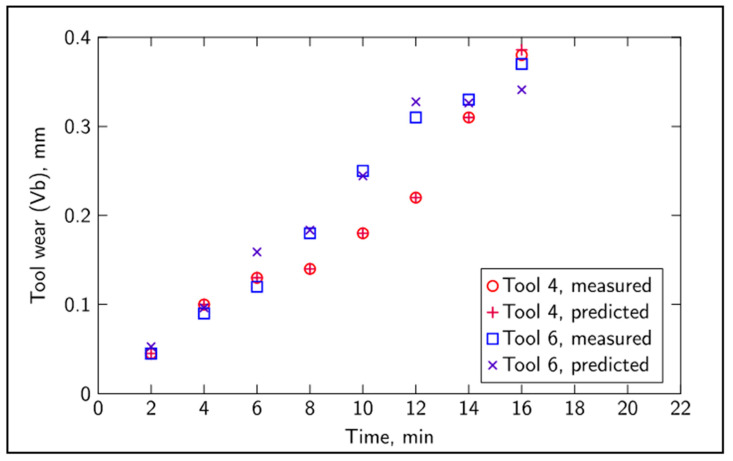
Classification performance on previously seen data (Tool 4) and unseen data (Tool 6).

**Figure 8 sensors-20-04493-f008:**
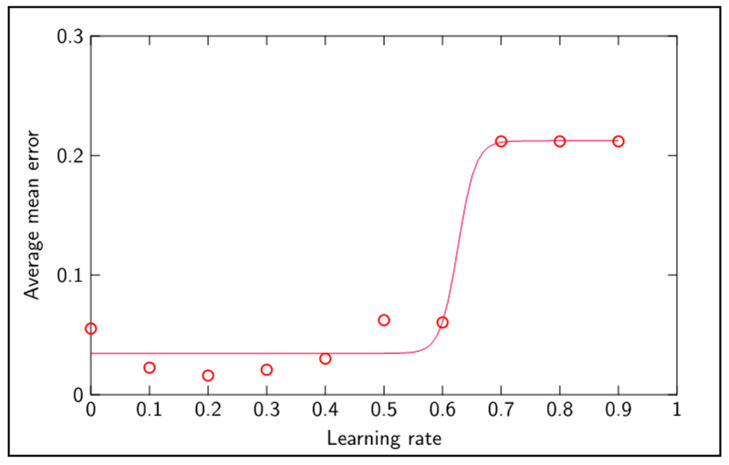
Average classification mean error versus learn rate.

**Figure 9 sensors-20-04493-f009:**
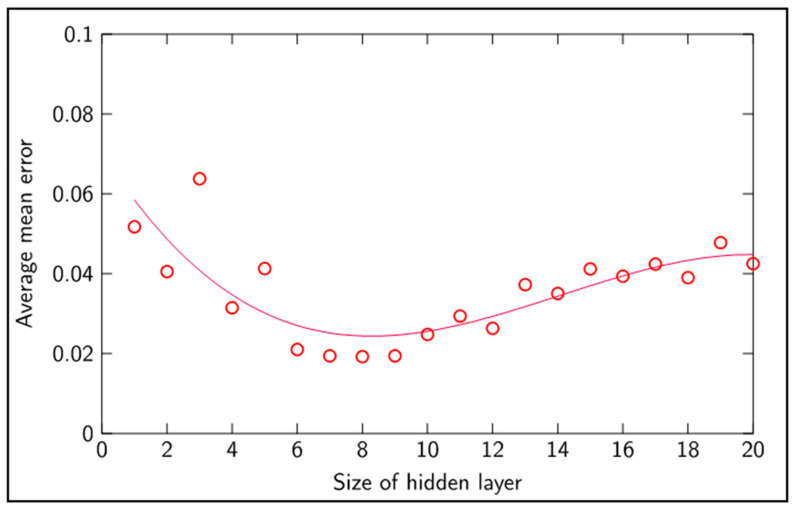
Average classification mean error versus size of hidden layer.

**Figure 10 sensors-20-04493-f010:**
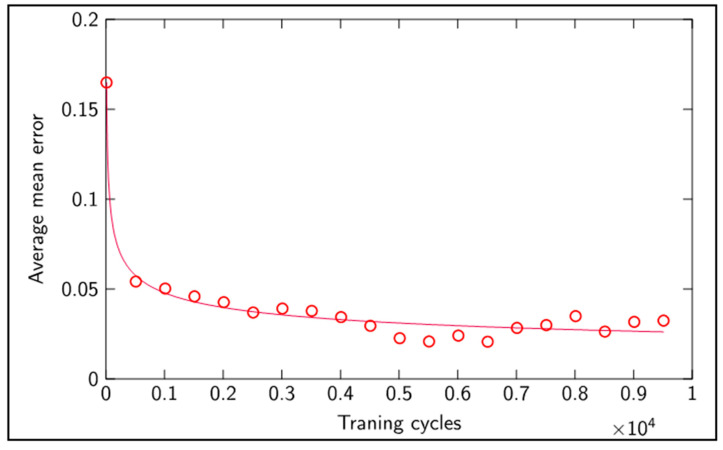
Average classification mean error versus number of training cycles.

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
