# Peer review of "A Novel Approach to Condition Monitoring of the Cutting Process Using Recurrent Neural Networks"

_sensors, 2020, doi:10.3390/s20164493_

Round 1
Reviewer 1 Report
In my opinion, the manuscript is well prepared, consistent and consistent. I read it with interest. The discussed issue is relevant in machining. My comments relate to minor mistakes: - instead of a modified lathe, a test stand based on a lathe should be used - the unit for the feed value is incorrect - which means: middle of the bar - line 174Author Response
Please see the attachment.

Reviewer 2 Report
The submitted manuscript proposes an approach to condition monitoring of the cutting process by multisensory to gather information from the cutting process using recurrent neural network (RNN). Feasibility and suitability have also been justified. The following suggestions are listed:
- The unit of cutting parameters should be checked carefully. For example, it says “… feed rate of 0.25 rev/min …” (Line 169, Page 4 of 11) but it seems to be not right. Spindle speed and/or original block (or cylindrical) workpiece of mild steel could be added.
- It is suggested that all the 14 features are listed to ease understanding of the analyzed features. It is not clear to know all those 14 features from the description – Line 176 – 179, Page 4 of 11 to 5 of 11.
- More explanation and information could be added for better illustration. For example, it says “The classification average mean error for Tool tip 4 (data used for training) was 0.0007 mm and the classification average mean error for Tool tip 6 (unseen dataset) was of 0.014 mm – results presented in Figure 7.” (Line 231 – 233, Page 6 of 11) However, it seems that there are no information for both “0.0007 mm” and “0.014 mm” in Figure 7!
- More experiments with various cutting parameters are suggested to be conducted to verify the proposed approach.
Round 2
Reviewer 2 Report
Authors have responded the reviewer’s comment.
With regard to the 14 features, the authors responded as the followings:
“A total of 14 features were extracted for each record at each of the tool wear levels – the extracted features for both sound and vibration, each accounting for 6 features, were: mean value, standard deviation, kurtosis, skewness and the power in the frequency bands 2.2–2.4 kHz and 4.4–4.6 kHz. Additionally, two more features were obtained from the mean value of the tangential and feed force.”
It means that both sound and vibration have six features. (2 * 6 = 12 features.) Additionally, two more features were obtained from the mean value of the tangential and feed force.” (2 features) Altogether 12 = 2 = 14 features.
But it says “ … each accounting for 6 features, were: mean value, standard deviation, kurtosis, skewness and the power in the frequency bands 2.2–2.4 kHz and 4.4–4.6 kHz.” Does it mean the six features are (1) mean value, (2) standard deviation, (3) kurtosis, (4) skewness, (5) power in the frequency bands 2.2–2.4 kHz, and (6) power in the frequency bands 2.2–2.4 kHz? If that’s the case, why two frequency bands were not considered for the first four - (1) mean value, (2) standard deviation, (3) kurtosis, (4) skewness?
Furthermore, the following information are suggested to be added for more persuasive and robust contribution of the submitted manuscript because the manuscript is entitled " ... cutting process ...".
- The spindle speed and the diameter of the cylindrical workpiece in “Figure 2 Experimental Apparatus” should be added.
- A few combinations of spindle speed and feed rate per revolution are suggested to be conducted for the verification of repeatability for the results from the proposed methods. It seems that the submitted manuscript only used one set of cutting parameters in the experiment. Although there are six tools adopted in the experiment, there is only on set of cutting parameters – “cutting speed is 350 mm/min”, “spindle speed is fixed and unknown”, “feed rate per revolution is fixed and unknown”.
